# Rhodophyta DNA Barcoding: Ribulose-1, 5-Bisphosphate Carboxylase Gene and Novel Universal Primers

**DOI:** 10.3390/ijms25010058

**Published:** 2023-12-19

**Authors:** Faith Masilive Mshiywa, Shelley Edwards, Graeme Bradley

**Affiliations:** 1Department of Biochemistry and Microbiology, University of Fort Hare, Alice 5700, South Africa; 2Department of Zoology & Entomology, Rhodes University, Makhanda 6139, South Africa; s.edwards@ru.ac.za

**Keywords:** DNA barcoding, *rbc*L, primers

## Abstract

Red algae (Rhodophyta) are a heterogeneous group of marine algal species that have served as a source of high-value molecules, including antioxidants and scaffolds, for novel drug development. However, it is challenging to identify Rhodophytes through morphological features alone, and in most instances, that has been the prevailing approach to identification. Consequently, this study undertook the identification of red algae species in Kenton-on-Sea, South Africa, as a baseline for future research on red algae biodiversity and conservation. The identification was achieved by designing, analysing, and using a set of universal primers through DNA barcoding of the *rbc*L gene. The PCR products of the *rbc*L gene were sequenced, and 96% of the amplicons were successfully sequenced from this set and matched with sequences on BOLD, which led to these species being molecularly described. Amongst these species are medicinally essential species, such as *Laurencia natalensis* and *Hypnea spinella*, and potential cryptic species. This calls for further investigation into the biodiversity of the studied region. Meanwhile, the availability of these primers will ease the identification process of red algae species from other coastal regions.

## 1. Introduction

Red algae, also known as Rhodophyta, are diverse groups of organisms of various species and are mostly found in oceans worldwide [1,2]. They possess large amounts of bioactive compounds with great medicinal value and have received considerable attention in recent years as antioxidants [3,4]. The South African coastline exhibits a diverse array of endemic red algae, possibly due to the unique environment created by the mixing of warm and cold waters [5]. As the decline in biodiversity is one of the most critical challenges of the 21st century, caused by the lack of knowledge on the state and distribution of biodiversity, it is crucial for the biodiversity of such commercially important species to be conserved [6].

Taxonomy provides an understanding of biodiversity that is a prerequisite for most biological research. However, traditional taxonomic methods that rely on morphological features alone are said to have contributed to the challenge at hand, especially in seaweeds [6,7]. This is due to cryptic or juvenile species having similar appearances that which make it difficult to differentiate between them [7]. Such species can only be detected with the use of molecular techniques [1,2].

One of the molecular methods that have been created to identify species like red algae with complex morphologies that make identification difficult is DNA barcoding. This was primarily introduced to increase the rate at which species are identified in response to the rate of biodiversity extinction [1,8]. This molecular technique involves sequencing a short barcode/gene to discriminate between species [9,10,11].

The plastid gene of the large subunit of ribulose-1, 5-bisphosphate carboxylase (*rbc*L) has been extensively used in barcoding studies and has been considered a promising barcode marker in red algae [1,12,13,14,15]. This is because, in contrast to plant mitochondrial and nuclear genes, the chloroplast genome has a basic, stable genetic structure and a low pace of evolution [16,17]. This enables sequencing and amplification even in degraded DNA [18].

Various primers have been designed for the *rbc*L gene; however, none of them are universal as there is no evidence of them amplifying red algae from different genera [14,19,20,21]. Therefore, there is a need for the development of universal primers for assessing red algal diversity at a rapid rate.

This study aimed to develop universal primers for the DNA barcoding of the *rbc*L gene to identify diverse red algae species from Kenton-on-Sea, South Africa.

## 2. Results

The newly designed universal primer set had sequence lengths between 18 and 24 base pairs for amplification of the *rbc*L gene in red algae (Table 1). The melting temperatures were 56 °C and 52 °C, and the GC contents were 40.0% and 44.4% for the forward and reverse primers, respectively. These parameters allowed for the annealing temperature of the primers to be determined, which was ~50 °C.

The study sequenced 26 red algae species, identified from Megablast [22] and the Barcode of Life Data System (BOLD) [23,24], with the use of the *rbc*L DNA barcode (Table 2; Figure 1). High similarities (>91% and up to 100%) with the species from MegaBlast and BOLD indicate high confidence in the identification of each specimen (Table 2).

Both phylogenetic tree construction methods produced congruent topologies, with the Bayesian Inference producing a better-resolved tree (Figure 2). The orders of the class Florideophyceae were monophyletic, as were the families, with one exception. The family Caulacanthaceae nested within Areschougiaceae, though this family was represented by only one sequence (7_Heringia mirabilis). Perhaps it was a misidentified species, or this family needs more attention from a taxonomic standpoint. The *rbc*L gene appears to be useful for delineating the species, though it appears that this is better achieved by using Bayesian Inference, rather than maximum likelihood algorithms. The deeper nodes between orders were not resolved and they formed a polytomy, though this is expected when using a faster-mutating organellar genomic gene. to construct phylogenetic trees.

Pairwise sequence divergences between orders ranged between 11.14% and 17.7%, intraorder/interfamily divergences values ranged between 11.9 and 17.7%, and intrafamily/intergeneric divergence was ~11% (Table 3). The family Caulacanthaceae, which nested within Areschougiaceae in the phylogeny, was 8.67% divergent from Areschougiaceae. The intrageneric sequence divergence values ranged between 4.2% and 9.2% (Table 4). The difference between the maximum intraspecific distance (6–6.5%) and the minimum interspecific distance (9–9.5%) (Figure 3) depicts the barcoding gap of this gene, which is 3%. This result validates the rbcL gene’s potential as a DNA barcode for red algae.

## 3. Discussion

### 3.1. Primer Design and Property Analysis

These primers (Table 1) were designed with careful adherence to acceptable primer properties in order to avoid poor amplification, sequencing issues, and mononucleotide repeats [25]. As a result, their sizes range from 18 to 24 bp because the primer length affects specificity and annealing [26]. The primers’ melting temperatures were set to between 52 and 60 °C to ensure efficient amplification; however, the G/C contents of these primers were slightly below the recommended range of 45% to 55%, which carries the disadvantage of a low melting temperature [26,27].

### 3.2. Red Algae Identification Based on Ribulose-1, 5-bisphosphate Carboxylase

Various species from different genera were identified from the *rbc*L gene, as shown in Table 2. Some species were found to have the same genetic makeup even though they had different morphological features; these were Samples 2 and K6 (Table 2 and Figure 1), which were both found to be Antithamnion pectinatu. This was also observed with samples 6 and 9 (*Delisea flaccida*); samples 4 and K22 (*Hypnea spinella*); samples 10, K13, and K27 (*Gelidium abbortiorum*);samples K9, K17, and K18 (*Ceramium obsoletum*); and samplesK19 and K20 (*Laurencia glomerata*). This phenomenon is called convergent morphology, which means that these species may differ in their phenotypical features because they are in different developmental stages. These developmental stages from juvenile to adult are influenced by different environmental factors such as temperature, nutrients, and light. Therefore, juveniles and adults do not always look alike but their nucleotide composition can be identical [28,29,30].

Identification went as far as the interspecific level (Table 3, Figure 1). The primers also allowed the identification of species within the same genera (intrageneric). However, samples K7, K8, and 8 were only resolved at the genus level, suggesting new or cryptic species. Therefore, further stringent taxonomic analyses would have to be conducted in order to name these “cryptic species”, as discovery through non-extensive DNA barcoding is controversial in nomenclature [31,32]. Additionally, the primer design process in this study showed the possibility of the *rbc*L gene not distinguishing between species thoroughly in different genera, leaving room for improvement.

The *rbc*L gene’s species-level discrimination ability is shown in Table 2 and Figure 2; however, as mentioned above, some species were only distinguished up to the genus level. The Bayesian Inference allowed for better resolution between clades closer to the tips but, due to the fast mutation rate of the *rbc*L gene, the deeper relationships (interorder relationships) were not well resolved. The genetic divergence of the *rbc*L gene is also said to account for the lack of precision in species-level identification [33]. However, in this study, pairwise distances for DNA barcoding analysis and a barcoding gap of 3% were calculated using the difference between the maximum intraspecific distance and the minimum interspecific distance, which was higher than the maximum intraspecies distance (Table 3 and Table 4). Figure 3 shows that there is no overlap in the range of intra-and interspecific *rbc*L sequence divergence; the existence of this apparent gap between these variations and the sizeable interspecific distance are properties that strengthen a marker as an ideal barcode region for its species discrimination ability. Therefore, the discrimination power of the *rbc*L barcode is considered valid; hence, correct species identification was enabled [34].

## 4. Materials and Methods

### 4.1. rbcL Sequence Alignment

Red algae *rbc*L sequences, *Ahnfeltiopsis glomerata* (AF388552), *Palisada perforata* (EU256330), *Ceramium pacificum* (FJ795539), and *Betaphycus gelatinus* (JX069190), from the Rhodophyta classes were obtained from GenBank (https://www.ncbi.nlm.nih.gov/ (accessed on 17 January 2017); [35]). A global alignment was performed on all the sequences using MEGA v6.0 [36]. This was performed by ClustalW where the pairwise alignment gap opening penalty was 15, and the gap extension penalty was 6.6. The same values applied to the multiple alignment gap opening penalty and the gap extension penalty. The DNA matrix used was ClustalW 1.6 with a transition weight of 0.5. The delay divergence cut-off percentage was 30. The alignments were inspected by eye for variable and conserved regions for designing primers. 

### 4.2. Universal Primer Design for Amplification of the Red Algae rbcL Gene

Universal primers, RFrbcLf1 and RFrbcLr2, designed from the sequence alignment were highly conserved to ensure that they would bind to as many red algae species as possible from these classes. The regions between the primer sets showed enough variability to ensure that the DNA barcode being developed was able to differentiate between different species of the Rhodophyta species. According to Kress and Erickson (2008), for a gene region to be considered a potential DNA barcode, it must contain significant species-level genetic variability and divergence. The analysis of the above sequence alignment showed that the rbcL primers met this criterion. The length of the amplified region was 977 bp in a conserved region with both the forward primer and the reverse primer. The reverse primer started at the beginning in the 5′-3′ direction at ATG to AAC. The reverse primer started from TTG to GTG in the 5′-3′ direction and a reverse complement was made.

Primer length was 18–20 base pairs, GC content was 40–60%, and G or C residues were added at the 3′ end. GC contents and melting temperatures were determined manually and confirmed using Gene Runner version 4.0.9.4 Beta [36].

### 4.3. rbcL Primer Universality Assessment for Barcoding Red Algae

The *rbc*L primers were assessed for universality using Megablast in GenBank for highly similar sequences (http://www.ncbi.nlm.nih.gov/BLAST (accessed on 11 February 2017); [22]). The several red algal families that emerged from this search for each primer are listed in Appendix A. The percentages of families and orders from this search are shown in Appendix A, respectively. Appendix A shows the sequences of the various primer pairs designed and Appendix A show that RFrbcLf1 and RFrbcLr2 was the best primer pair compared to the other primer sets.

### 4.4. Sample Collection

A total number of 31 red algae samples, about 50 mg each (of which 5 mg per species was used for the study) was collected in South Africa (33.6806° S, 26.6701° E) in 2016 and 2017 during low tide. They were rinsed with distilled water and stored at −80 °C.

### 4.5. Amplification and Sequencing of Red Algae Ribulose-1, 5-bisphosphate Carboxylase (rbcL) Gene

Genomic DNA from these specimens was isolated using the ZR Plant/Seed DNA MiniPrep Kit (Zymo Research Corporation, Irvine, CA, USA) and concentrations were measured using Nanodrop 2000 (Thermo Fisher Scientific Corporation, Waltham, MA, USA). A segment of 977 bp of the *rbc*L gene was amplified using PCR from the genomic DNA with RFrbcLf1 (5′GTCTAACTCTGTAGAAGAAC 3′) and RFrbcLr2 (5′GTCTAACTCTGTAGAAGAAC 3′) (Inqaba Biotechnica Industries (Pty), Muckleneuk, South Africa). PCR reactions were performed in 30 µL reactions for each sample. Each PCR tube contained 12.5 µL of the Promega Go Taq^®^ Green master mix (Promega Coporation, Madison, WI, USA), 2.5 µL of the forward primer (10 µM), 2.5 µL of the reverse primer (10 µM), 5 µL of the template DNA, and 7.5 µL of nuclease-free water. The reactions were carried out using the Bio-Rad My Cycler^®^ Thermal cycler (Bio-Rad Laboratories, Hercules, CA, USA). PCR reactions were performed with 1 cycle of 95 °C (initial denaturation) for 3 min; 40 cycles of 95 °C (denaturation) for 1 min; 50 °C (annealing temperature) for 30 s, 72 °C (extension) for 1 min, and 1 cycle of 72 °C (final extension) for 5 min.

Amplicons were sequenced with the PCR primers, analysed on a 3500 Genetic Analyzer (Thermo Fisher Scientific Corporation, Waltham, MA, USA), and assessed on Chromas 2.6.4 (Technelysium Pty Ltd., South Brisbane, Australia).

### 4.6. Species Identification Using DNA Barcoding and Phylogenetic Analyses

The sequences were used to identify species in Megablast on NCBI [37] and BOLD. The species name with the highest percentage of similarity from one of the search engines was used for the results. Two phylogenetic trees from 750 bp of the aligned sequences were constructed. The first was produced using the MEGA (Molecular Evolutionary Genetics Analysis) maximum likelihood algorithm, with the GTR (discrete gamma categories with invariant sites) evolutionary model chosen using AIC (Akaike information criterion). Branch supports were estimated using 1000 bootstrap replications, and 6 G + I categories were selected. The ML heuristic method was “Nearest-Neighbor-Interchange”, with a strong branch filter and one thread for system resource usage. The second phylogenetic tree was constructed using Bayesian Inference in MrBayes v.3.2.7 [38]. Priors in MrBayes were set according to the evolutionary model found using MEGA (GTR + Γ + I), and uniform priors were kept for all other parameters. The MCMC was run with two parallel runs for 10 million generations, with trees sampled every 1000 generations. The effective sample sizes (ESS) of all parameters, viewed in Tracer v.1.6.0 [39], were >200, so the number of generations to discard as burn-in was around 20%. A 50% majority rule tree was constructed, excluding the burn-in, using the ‘sumt’ command in MrBayes, and nodes with ≥0.95 posterior probability were considered supported. These two phylogenetic trees were used instead of Neighbour joining (NJ) because NJ is a clustering algorithm and models of substitution cannot be applied to the algorithm. It is more often that we find ML and Bayesian Inference presented in the modern literature [40].

### 4.7. Pairwise Genetic Distance Analysis

Sequence divergences were determined by estimating the uncorrected p-distances between and within species using MEGA, and the averages of interspecific and intraspecific distances were calculated in Species Identifier [41].

## 5. Conclusions

Universal primers for the *rbc*L gene and the DNA barcoding for a variety of red algae species in South Africa were designed and they were tested on species from Kenton-on- Sea to assess biodiversity. In total, 96% of those species were successfully sequenced and identified. Some of these species include those rich in polysaccharides, antioxidants, and cytotoxic compounds, like *Hypnea spinella*, *Plocamium corallorhiza*, *Delisea flaccida*, and *Gelidium pristoides* [42,43,44]. Potential cryptic species were identified, and this calls for further investigation of the biodiversity in that geographical area. Meanwhile, the availability of these primers will ease the identification process of red algae species from other coastal regions, and the sequences and identified species can be phylogenetically analysed. Ultimately, they can be uploaded to biological databases such as BOLD and GenBank.

## Figures and Tables

**Figure 1 ijms-25-00058-f001:**
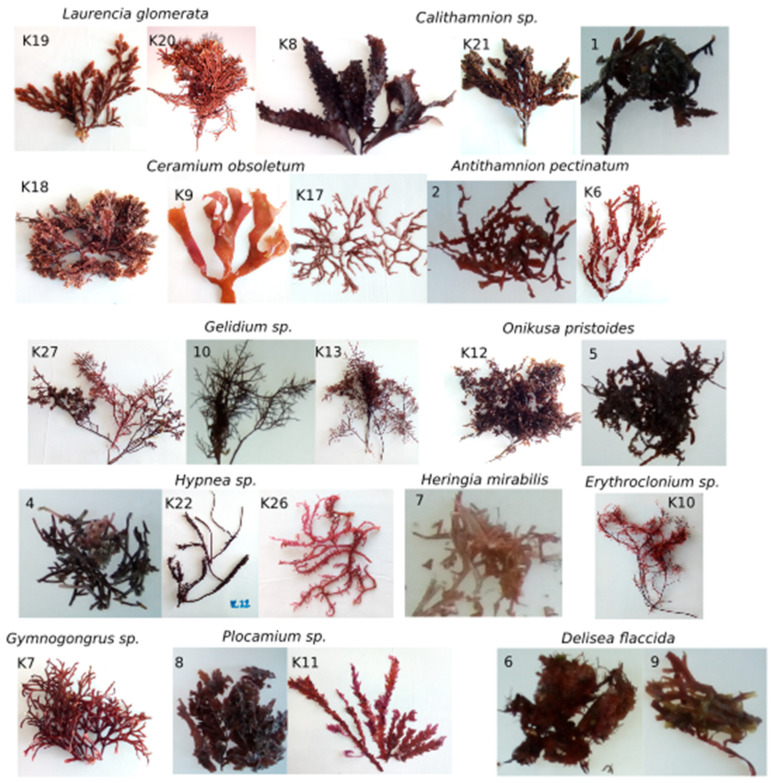
Photographs of the specimens sequenced in this study. Specimen numbers as in Table 2.

**Figure 2 ijms-25-00058-f002:**
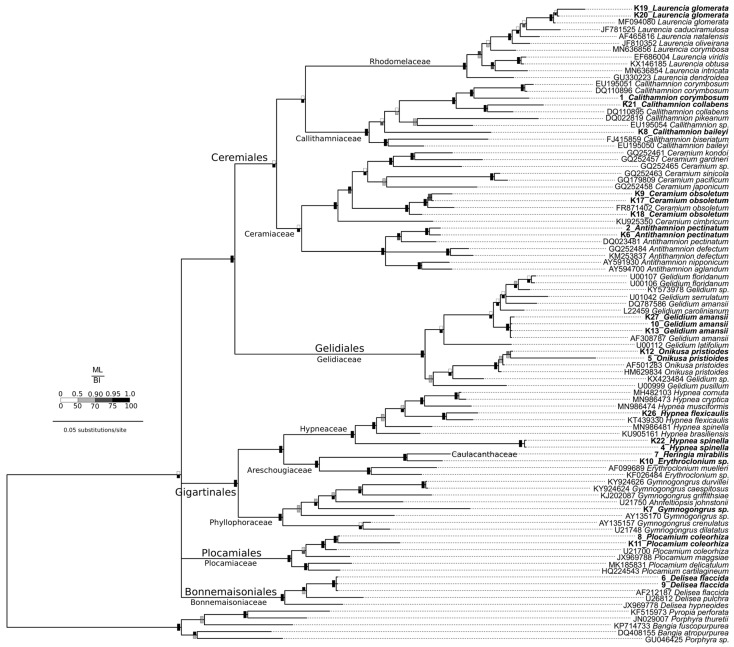
Bayesian Inference (BI) phylogenetic tree produced from *rbc*L sequence data from Algoa Bay red algae and GenBank sequences detected by the primers from Class Florideophyceae. Key to the support values shown as squares at each node, with the inset key (ML bootstrap percentages above the node, BI posterior probabilities below the node). Species in bold are the ones that were collected for this study.

**Figure 3 ijms-25-00058-f003:**
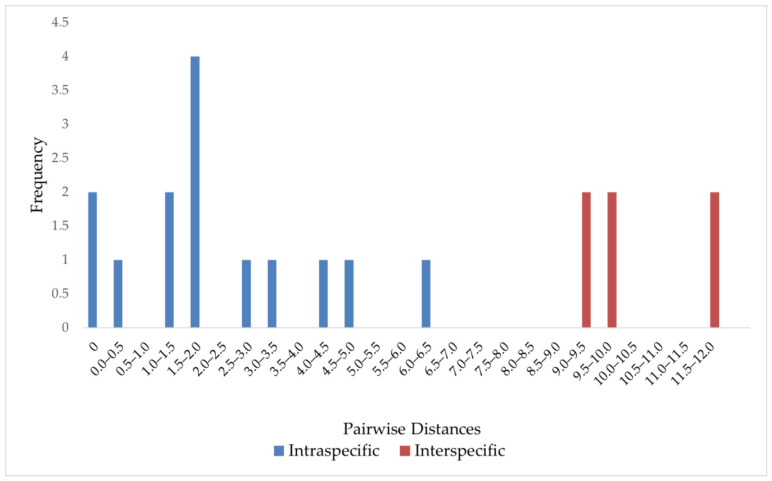
Frequency distribution of intraspecific and interspecific (congeneric) genetic divergence in red algae.

**Table 1 ijms-25-00058-t001:** Designed forward (f) and reverse (r) primers for *rbc*L region in Rhodophyta (R): Class Florideophyceae (F) for DNA barcoding, primer sequence, primer length, melting temperature (Tm), and GC content are shown.

Primer	Primer Sequence (5′-3′)	Primer Length (bp)	Tm (°C)	GC%
*RFrbc*Lf1	GTCTAACTCTGTAGAAGAAC	20	56	40.0
*RFrbc*Lr2	GCCCAATCTTGTTCAAAG	18	52	44.4

**Table 2 ijms-25-00058-t002:** DNA barcode-verified specimen identities of red algae from Kenton-on-Sea. The sequence IDs, similarity percentages, and the collection data are shown.

Sample ID	Barcode of Life Data (Top%Speciesid)	E-Value	Collection Data			
			Location	Geographic Coordinates	Date	Genbank Accession Numbers
1	*Callithamnion corymbosum* (93.50)	0.0	South Africa	33.6806° S, 26.6701° E	10 September 2016	OR939833
2	*Antithamnion pectinatum* (96.90)	0.0	South Africa	33.6806° S, 26.6701° E	10 September 2016	OR939834
4	*Hypnea spinella* (91.80)	0.0	South Africa	33.6806° S, 26.6701° E	10 September 2016	OR939835
5	*Gelidium pristoides* (95.10)	0.0	South Africa	33.6806° S, 26.6701° E	10 September 2016	OR939836
6	*Delisea flaccida* (100.00)	0.0	South Africa	33.6806° S, 26.6701° E	10 September 2016	OR939837
7	*Erythroclonium angustatum* (90.50)	0.0	South Africa	33.6806° S, 26.6701° E	10 September 2016	OR939838
8	*Plocamium coleorhiza* (99.70)	0.0	South Africa	33.6806° S, 26.6701° E	10 September 2016	OR939839
9	*Delisea flaccida* (99.90)	0.0	South Africa	33.6806° S, 26.6701° E	10 September 2016	OR939840
10	*Gelidium amansii* (100.00)	0.0	South Africa	33.6806° S, 26.6701° E	10 September 2016	OR939841
K6	*Antithamnion pectinatum* (96.80)	0.0	South Africa	33.6806° S, 26.6701° E	24 November 2017	OR939842
K7	*Gymnogongrus* sp. (92.30)	0.0	South Africa	33.6806° S, 26.6701° E	24 November 2017	OR939843
K8	*Callithamnion bailey* (92.80)	0.0	South Africa	33.6806° S, 26.6701° E	24 November 2017	OR939844
K9	*Ceramium obsoletum* (97.20)	0.0	South Africa	33.6806° S, 26.6701° E	24 November 2017	OR939845
K10	*Erythroclonium angustatum* (91.50)	0.0	South Africa	33.6806° S, 26.6701° E	24 November 2017	OR939846
K11	*Plocamium coleorhiza* (96.30)	0.0	South Africa	33.6806° S, 26.6701° E	24 November 2017	OR939847
K12	*Gelidium pristoides* (99.70)	0.0	South Africa	33.6806° S, 26.6701° E	24 November 2017	OR939848
K13	*Gelidium amansii* (100.00)	0.0	South Africa	33.6806° S, 26.6701° E	24 November 2017	OR939849
K17	*Ceramium obsoletum* (98.40)	0.0	South Africa	33.6806° S, 26.6701° E	24 November 2017	OR939850
K18	*Ceramium obsoletum* (96.70)	0.0	South Africa	33.6806° S, 26.6701° E	24 November 2017	OR939851
K19	*Laurencia glomerata* (98.24)	0.0	South Africa	33.6806° S, 26.6701° E	24 November 2017	OR939852
K20	*Laurencia glomerata* (99.686.20)	0.0	South Africa	33.6806° S, 26.6701° E	24 November 2017	OR939853
K21	*Callithamnion collabens* (95.50)	0.0	South Africa	33.6806° S, 26.6701° E	24 November 2017	OR939854
K22	*Hypnea spinella* (93.90)	0.0	South Africa	33.6806° S, 26.6701° E	24 November 2017	OR939855
K26	*Hypnea flexicaulis* (97.60)	0.0	South Africa	33.6806° S, 26.6701° E	24 November 2017	OR939856
K27	*Gymnogongrus* sp. (99.90)	0.0	South Africa	33.6806° S, 26.6701° E	24 November 2017	OR939857

**Table 3 ijms-25-00058-t003:** Sequence divergence values between genera of Rhodophyta (given as percentages).

	Order	Family	Genus	1	2	3	4	5	6	7	8	9	10	11
1	Ceremiales	Rhodomeiaceae	*Laurencia*											
2	Ceremiales	Callithamniaceae	*Calithamnion*	14.72										
3	Ceremiales	Ceramiaceae	*Ceramium*	13.52	12.46									
4	Gelidiales	Ceramiaceae	*Antithamnion*	13.32	12.48	11.14								
5	Gigartinales	Gelidiaceae	*Gelidium*	17.23	16.15	15.12	15.98							
6	Gigartinales	Hypneaceae	*Hypnea*	17.95	16.60	16.42	16.49	16.65						
7	Gigartinales	Areschougiaceae	*Erythroclonium*	17.16	16.15	15.93	16.14	17.63	11.94					
8	Gigartinales	Caulacanthaceae	*Heringia*	17.36	15.68	15.74	16.66	17.16	12.64	8.67				
9	Gigartinales	Phyllophoraceae	*Gymnogongrus*	17.44	15.66	16.09	15.98	17.68	14.41	13.26	14.35			
10	Plocamiales	Plocamiaceae	*Plocamium*	17.71	15.88	14.50	15.46	16.97	15.87	15.08	14.18	15.23		
11	Bonnemaisoniales	Bonnemaisoniaceae	*Delisea*	15.55	15.43	14.41	14.13	16.77	15.60	15.70	15.73	14.30	12.54	
12			Outgroup	20.75	19.47	19.14	19.64	20.47	19.55	19.46	19.17	20.00	19.77	19.40

**Table 4 ijms-25-00058-t004:** Intrageneric sequence divergence within genera of Rhodophyta (given as percentages) and the standard error of the mean are shown.

	%Distance	SE
*Calithamnion*	8.67	0.62
*Antithamnion*	4.86	0.55
*Gelidium*	5.90	0.50
*Heringia*	9.05	1.05
*Delisea*	3.74	0.44
*Ceramium*	7.72	0.62
*Plocamium*	5.15	0.51
*Erythroclonium*	8.29	0.81
*Laurencia*	4.27	0.43
*Hypnea*	7.32	0.56
*Gymnogongrus*	9.20	0.71
Outgroup	10.11	0.74

## Data Availability

The data presented in this study are openly available in GenBank (www.ncbi.nlm.nih.gov) (See Table 2 for accession numbers).

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
