# Peer review of "Rhodophyta DNA Barcoding: Ribulose-1, 5-Bisphosphate Carboxylase Gene and Novel Universal Primers"

_ijms, 2023, doi:10.3390/ijms25010058_

Round 1

Reviewer 1 Report

Comments and Suggestions for Authors

The authors present the development of the novel universal primer for DNA barcoding. 

This is good for the situations which obtaining the sequence for identification is failed. 

However, as its importance is high, the methods are strictly designed. 

Although the result and the discussion are important in this kind of the research, the M&M should be also followed strictly. 

In my opinion, the M&M section should be improved, indicating the strict information such as the species number, and individuals number. 

Additionally, the identification should be double checked, as the sequences in the database (e.g., BOLD, NCBI) are sometimes erronously uploaded by misidentification. 

I would suggest that the authors verify the species identification from the morphological taxonomist, making the result (their primers) more reliable.

In addition, 

as far as I know, the NJ analysis is more common in the barcoding study than the ML analysis, as the mechanism for analysis. 

I would add the NJ analysis and compare them in order to upgrade the reliance.  

Please check the minor comments in the attachment. 

Comments on the Quality of English Language

The senteces are not easy to understand and are too long.

Several paragraphs and sentences need the logical writing.

Overall English correction and minor checking for writing (e.g., species name should be italic) are needed.

Author Response

All of the other comments have been addressed:

  1. Line 30-33:

I understand what you mean, but it would be better if this sentence is more concise.

Response/Modification: The statement has been made concise as recommended by the reviewer. The correction will be seen on page 30-33.

  1. Line 33-35:

I understand also what you mean, but the logic is weak.

Response/Modification: The statement has been modified as seen in lines 33-35.

  1. Line 37-39:

This sentence is too aggressive. I know that the authors drive their justification for their study. However, modest expression would be better. The statement has been changed as seen in lines 37-39.

  1. Line 123:

As far as I know, the NJ analysis is often used for Barcode study. It would be better if the authors made both topologies and compared them.

Response/Modification: We have a different opinion on this one because according to one of our authors (SE); NJ is no longer really an acceptable method when using molecular data, because of the type of data used. NJ is a clustering algorithm, and models of substitution cannot be applied to the algorithm. It is more often that we see ML and Bayesian Inference presented in modern literature. They therefore suggest we do a Bayesian Inference a second phylogeny and for comparison. Please see the reference link below:

https://www.researchgate.net/post/Whats-the-difference-between-neighbor-joining-maximum-likelihood-maximum-parsimony-and-Bayesian-inference

The insertion of about the construction of the Bayesian Inference tree is seen in line 24 to line 250. Justification in lines l63 to line 166. New figures inserted in 78, 93.

  1. Line 153:

should be italic in a species name.

Response/Modification: Species names have now been changed to italics as seen in line 143.

  1. Line 204:

How many species and the individuals per species used in this study?

Response/Modification: Species numbers added as seen in Line 212.

Please see the attached manuscript and responses to reviewers.

Warm regards.

Reviewer 2 Report

Comments and Suggestions for Authors

Research topic about Rhodophyta DNA barcoding represents a very interesting and important topic but the authors did not cover the topic with their research results in an appropriate way and there are many questions and unresolved issued in all the manuscript chapters. Introduction could be longer and more extensive. Methods have some questionable parts that significantly influence the soundness of the research. Results were written only as descriptions of the figures. Literature citations are also questionable. I advise general revision of the manuscript by the authors. 

Author Response

All of the reviewer's comments have been addressed.

  1. Line 71

Please give reference to the database used here. It is very unusual that BOLD was used since it gives a collection of COI reference sequences and not rbcL. Please if authors could explain this.....

Response: BOLD was earlier used for only CO1 references; However, the article below states that it was initially said that other single-gene or multi-gene can be supported by BOLD, and that includes rbcL and MatK for plant identification and ITS for fungal identification.

“Currently, bold is exclusively populated with COI data, but it can support other single-gene or multigenic barcodes. As a result, it is positioned to deal with the additional data storage requirements created when supplemental barcode regions gain registration for the animal kingdom or as alternate barcode regions are designated for the other kingdoms of life”.

 Ratnasingham, S. and Hebert, P.D., 2007. BOLD: The Barcode of Life Data System (http://www. barcodinglife. org). Molecular ecology notes7(3), pp.355-364.

As a result, the Consortium for the Barcode of Life (CBOL) approved two plastid loci, rbcL and matK, as the official DNA barcode for all land plants, while urging further data PLoS ONE | www.plosone.org 1 October 2011 | Volume 6 | Issue 10 | e26597 collection on trnH-psbA to assess its potential to be added to the land plant barcode

All the added depositories are readily available now in the updated version of BOLD; BOLD systems V4, see link below with the above-mentioned depositories:

https://www.boldsystems.org/index.php/IDS_OpenIdEngine

             Another article explains this phenomenon further:

Hollingsworth, P.M., Graham, S.W. and Little, D.P., 2011. Choosing and using a plant DNA barcode. PloS one6(5), p.e19254.

Citations included as seen in line 71and in the reference list.

  1. Line 78: unusual that all E-values were exactly zero...please authors to comment and check in the results:

Response/Modification: Results were compared to BLAST N (Megablast) with highest similarity and the e-values were zero. The species name with the highest similarity between BOLD and BLAST was the one used for the study. Please find the sequences from this study that were submitted in both search engines and will be accessioned in Genbank, see Table 1 (T.B.A." (To Be Accessioned).

Citations added in line 71 to 74 and line 225 to 227.

  1. Line 80: divergence between genera?

         Please explain better this part....

Response/Modification: New table in line 120. Section rewritten for clarity in line 103 to 111.

  1. Line 83 to 84: Sentence not finished??

Response/Modification: Sentence deleted

  1. Line 149:

Make-up?

Response/Modification: Word changed to makeup in line 139.

  1. Line 186:

         Only four reference sequences were use for an alignment.

Response/Modification: Yes, this number was reduced from 95 sequences because of the lack of conserved reserved regions in multiple sequences across Classes of Rhodophyta. These four sequences represented many species in the Florideophyceae Class as their conserved domains aligned/matched with several species.

  1. Line 189: Give details about the parameters used for the alignment.

Response/Modification: Details given as seen in line 179.

  1. Line 193: Give length and position of the amplified region with this new designed primer pair:

Response/Modification from 190 to 200. Length and position added as seen in line 187 to 197.

  1. Line 201: supplementary data is not explained well primers in Table S1 were not all explained in the text?

Response/Modification: Table S1 explained in 205 to 207.

  1. Line 227: Unclear what was the control (positive?) and what was the amplified samples. Was the size of the amplified fragment 1400 bp? This is very long for a barcode. How was this Sanger sequenced since read length of Genetic analyser is ≥420 bp?

Response/Modification: Sentence has been deleted, and PCR reaction conditions added in line 219 to 226

  1. Line 216: BOLD

Response/Modification: Mentioned in Line 71: Barcode of life data systems.

  1. Line 226: What was the length of the alignment used for tree construction?

Response/Modification: The length of the alignment has been mentioned in line 232.

  1. Line 226: Is this the correct citation for the program used?

Response/Modification: The citation has been corrected.

 Please see the attached document with the manuscript and responses to reviewers.

Warm regards

Round 2

Reviewer 1 Report

Comments and Suggestions for Authors

The authors addressed most of the issues suggested.

The only one thing I would suggest is that the authors indicate the reason not to use NJ in 4.6 section simply, based on the response with the reference. 

It will provide the information for general readers. 

Author Response

Dear Reviewer 1,

Please find the attached document with the response to the second round of reviews.

Thank you.

Reviewer 2 Report

Comments and Suggestions for Authors

Corrections have been made according to suggestions. 

Author Response

Dear Reviewer 2,

Thank you for your comment.